# The Perception of Workplace Safety and of Risk of Contagion among Preschool Teachers during the COVID-19 Pandemic

**DOI:** 10.3390/children10071222

**Published:** 2023-07-14

**Authors:** Giulia Bacci, Daniela Converso, Ilaria Sottimano, Mara Martini

**Affiliations:** Department of Psychology, University of Turin, 10124 Turin, Italy; giulia.bacci@unito.it (G.B.); ilaria.sottimano@unito.it (I.S.); mara.martini@unito.it (M.M.)

**Keywords:** COVID-19, early childhood education, workplace safety, preschool teachers, children, fear of infection

## Abstract

The COVID-19 pandemic had a major impact on early childhood educational contexts and on educators’ working conditions. This study aims to examine the change over time in personal contribution to workplace safety and perception of risk of infection among preschool teachers after returning to in-person work during the third wave of the pandemic (spring 2021). Teachers’ perceptions of workplace safety can influence their quality of work-life and, as a consequence, the quality of service offered to children. Data were collected using two questionnaires: at T0 (January) and at T1 (May). The results showed relations between organizational and personal actions to manage risk and fear of infection at work. Concern about COVID-19 contagion decreased over time (*t* = 5.53, *p* = 0.000) and perceptions of personal contribution to workplace safety related to COVID-19 improved: *t* = −2.18 *p* = 0.031. The decrease in these concerns illustrates how perceptions of ability to manage contagion and protect workplace safety of preschool teachers improved over time, despite the stability of the pandemic context in the first half of 2021. After the end of the COVID-19 pandemic, this study gives an account of some good practices and their perceived effectiveness in terms of safety for childhood educational contexts.

## 1. Introduction

The COVID-19 pandemic had significant implications for the work environment from both organizational and social perspectives. Several studies have focused primarily on health care workers without considering other categories of workers who were at the forefront of COVID-19 infection risk because of their work that frequently requires them to interact with the public [1,2,3,4,5,6], who can both carry infection and be infected. Among these categories of workers, it is important to highlight kindergarten and preschool teachers (kindergarten teachers take care of children who are from 0 to 3 years old; preschool teachers teach children aged 3 to 6), who have experienced a unique situation in terms of infection risk. Educators and preschool teachers were expected to fulfil their educational duties while following guidelines to contain the virus [7]. As with other workers, social contact plays a large role in teachers’ work, as it often requires physical proximity to students and children. Although children could also be taught good hygiene practices to help contain the spread of the epidemic, it was much more difficult to establish new routines and ensure that children follow them [8]. This exposed both professionals and children to a higher risk of contagion.

In addition, the use of masks was not required for children under six years of age, and protective devices for adults must not interfere with the ability to be detected and to maintain close contact with and between young children [9]. Neither at the time these services were first reopened (May–June 2020), when vaccination for COVID-19 was not available for workers, nor today, after the vaccination campaign for much of the adult population, did this potential risk of infection change. Vaccination was not, and still is not, available for children under six years of age. In this setting, even for vaccinated workers, the concern about risk of infection could continue longer. Nevertheless, to our knowledge, there is limited research addressing the experience of this particular type of worker during the pandemic [10,11], especially in Italy, where this population seems to be quite neglected. A study conducted in a different context (Denmark) showed that in a sample of 174 teachers, 43% of participants reported feeling nervous or anxious about the outbreak of COVID-19 [11]. In this study, most reported fear of infection or fear of transmitting the infection from work to home, the same as employees working in elder care, child-care, hospitals and rehabilitation facilities [12]. Several studies [13,14] have shown that one of the main causes of healthcare workers’ fears during the pandemic was the fear of infecting family members, which was even greater than the fear of infecting themselves. In addition, another study showed that the prevalence of teachers who feared spreading the infection to their students was comparable to the numbers observed for childcare workers and lower than those observed for elder care workers [7]. Several studies have made it clear that these fears regarding the risk of infection reduce wellbeing in the workplace [10,13,15] and thus reduce workers’ composure, which should be important in providing a positive and safe service to 0–6-year-old children.

Importantly, the appropriateness of school closures as a means of controlling the spread of COVID-19 was questionable, especially since there was evidence of the limited effect of school closures on the spread of the virus. In Italy, the Scientific and Technical Committee (STC) of the Istituto Superiore di Sanità (ISS; National Institute of Health) developed guidelines for reopening schools at all levels on open grounds starting in May 2020 [16]. The municipality of the major city in northern Italy involved in this study opened the gardens of public kindergartens to support families in the presence of educators, to restore forms of normality and relief for families by allowing them to go outside, to be in a familiar and safe place for children and to meet with teachers and educators while maintaining distance and avoiding gatherings (http://www.comune.torino.it/ucstampa/2020/article_381.shtml (accessed on 1 July 2023)).

However, in relation to early childhood educational institutions, given the need for physical contact that characterizes the relationship between children and their peer group and adult caregivers, as well as aspects of care for children by educators and support staff, it has been necessary to introduce organizational methods that consider the difficulty of ensuring physical distance, if not between adults. The relationship between children and adults is the prerequisite for giving meaning to children’s attendance at an educational institution, which should be characterized as a highly affective social experience. Due to the unstable spread of the pandemic, several decrees were issued in Italy from March 2020 to regulate various aspects of social and productive life to reduce the spread of contagion. The decree of 3 August 2020, focused on the reopening of educational and preschool services, created the so-called “bubble system” for educational services for 0–6-year-olds, which provided for the creation of small stable groups to which one or more educators and staff were assigned in order to contain the spread of contagion.

The organization of the spaces provides for areas arranged according to the needs of each age group, with even the furniture arranged differently, so that the proposed daily experiences could be carried out respecting the principle of non-overlap between different groups, using play and learning materials, objects and toys exclusively assigned to specific groups/departments. The decree also provided for the use of outdoor areas and their transformation to accommodate permanent relationship and play groups [9]. All these requirements for the organization of educational services for 0–6-year-old children were aimed at protecting both educators/teachers and children (with their families) from the risk of infection. This type of intervention seemed to be important to improve wellbeing at work and to reduce emotional exhaustion [15], although to our knowledge this aspect has not yet been analyzed in the specific context of Italian educational services. Based on these premises, our study first focused on perceptions of safety and risk among employees of the education system of a large city in northern Italy, considering both the organizational precautions taken to manage the COVID-19 risk of contagion and the personal contribution of employees to the management of COVID-19 risk in the workplace. In this direction, we considered risk perception as subjective assessment of the probability of occurrence of a certain type of accident and how concerned we are about such an event [1]. According to Falco et al. [15], the perceived risk of contracting COVID-19 at work could be considered a work demand (which could increase work-related stress); in contrast, the organization’s ability to address the risk of infection and personal contribution can be considered resources. Concerns about the risk of contagion in the workplace and about the particular characteristics of its users (children under six) could still be high among this type of education provider. It is important to find ways to reduce this source of work-related stress to improve their wellbeing at work and the resulting impact on the quality of educational services. Therefore, the aim of the present study was to measure the educators’ workplace concerns and their perception of safety at work in relation both to organization ability to provide safe workplace conditions and to the ability of educators themselves to contribute to safe working conditions, during the second phase of the COVID-19 pandemic. This ability may fall under the so-called “non-technical skills” (NTS) [15,17], which can be defined as “cognitive, social and personal skills, complementary to technical skills, which contribute to safe and efficient performance” [17], (p. 2). The perceptions of the ability to manage contagion and protect workplace safety of preschool teachers could contribute to decreasing their concern about risk of contagion, as we will see. Although the COVID-19 pandemic is now ended, having tools and practices that enable successful work-related responses as well as supporting the development of NTS to enhance self-efficacy perception and wellbeing can be important in the case of possible new pandemics [18,19].

## 2. Materials and Methods

### 2.1. Measures

The questionnaire included questions exploring the feeling of safety at work during the period of COVID-19. We used the SAPH@W (Safety at Work) questionnaire, which focuses on the specific risk of COVID-19 infection and has been validated in the Italian context with different types of workers facing the pandemic in their organizational setting [20]. The instrument, inspired by the NTS perspective [17] and designed to assess workers’ perceptions of safety in organizations, consisted of 20 items (4 in each subscale) with a five-point response scale (from 1 “not at all” to 5 “completely”) exploring five content areas of perceptions of workplace safety during the COVID-19 pandemic. Most items addressed perceptions of safety among workers resulting from the organizational response to the pandemic, whereas a few items were devoted to measuring workers’ personal contributions to managing COVID-19 risk at work.

Specifically, SAPH@W consists of five subscales:Communication: the scale investigates the efficacy and effectiveness of the organizational communication on COVID-19 (for example: “In your opinion, in your workplace there is the opportunity to communicate effectively with the supervisor on risks related to COVID-19”). The Cronbach’s alpha was 0.87.Decision making: this scale investigates promptness, foresight and care of the organizational decision-making process regarding COVID-19 (for example: “Regarding contagion risks by COVID-19, in your opinion, your employer organization is able to make quick decisions”). The Cronbach’s alpha was 0.95.Situational awareness of contagion risks: the scale focuses on the perceived capability of managers and security staff to adequately monitor contagion risks. Item example: “In your opinion, your employer organization is able to identify specific contagion risks by COVID-19 in your job”. The Cronbach’s alpha was 0.93.Fatigue management: investigates the organizational ability to recognize and care for workers’ fatigue specifically due to the pandemic. Example of item: “In order to contain contagion risks by COVID-19, it is important that each employee adopts specific behaviors (i.e., using PPE, keeping interpersonal distance, practicing remote working). In your opinion, your employer organization can recognize the possible effects of such behaviors on physical fatigue”. The Cronbach’s alpha was 0.94.Personal contribution to workplace safety in relation to COVID-19 regarding the four dimensions cited above. Item example: “Think about your feelings regarding your job in this phase. Do you feel able to provide information to other employees regarding how to tackle contagion risks by COVID-19”. The Cronbach’s alpha was 0.87.

An ad hoc scale of concern for COVID-19 infection in the workplace with a 10-point answer scale (from 1 “Not at all” to 10 “Completely”) was also included, which investigated concern about contagion. The focus was on fears of contagion in the workplace (Items: 1. How worried are you about the possibility of being infected in the workplace?; 2. How likely are you to be infected with COVID-19 in workplace?; 3. How worried are you about the possibility of being infected in everyday life?). The Cronbach’s alpha was 0.86.

### 2.2. Participants

The participants were identified during an online training course, during which they were asked to participate in the study and their email addresses were collected. The questionnaire was sent to 410 preschool teachers in kindergartens and preschools in the municipality. A total of 108 of these staff responded to both the first (T0) and second (T1) questionnaires. All participants were female, as the prevalence of women in educational work is typical of the Italian context. The average age was 49.2 years. A total of 81% of participants live with husbands and 70.4% of participants have children. A total of 12% of teachers have elderly relatives.

### 2.3. Data Collection

Data were collected based on two self-report questionnaires administered to preschool teachers in a large northern Italian city. The first questionnaire was sent in January 2021 (T0), and the second in May 2021 (T1). In January, the RT contagion index in Piedmont was about one at the lower boundary, consistent with a Type 2 scenario. On 11 January 2021, this area was considered a “yellow zone”, meaning that the following were in effect: travel bans between regions and autonomous provinces, a 10 p.m. curfew, many activities were subject to restrictions (reduced opening hours, closure of shopping centers on holidays except for stores with basic needs), many activities were closed (gyms, swimming pools, etc.), and remote work was strongly recommended for offices. Despite this situation, schools were open, especially kindergartens and preschools, so that educators and preschool teachers had to work in attendance. In May 2021, during the second survey, Piedmont was still in the yellow zone; however, there were signs of improvement in the number of infections (RT index just below one). Both questionnaires were sent by e-mail. Workers’ participation was completely voluntary, and anonymity of data collection was guaranteed by the research group of the Department of Psychology. The survey was in accordance with the 1995 Declaration of Helsinki, which was revised in Edinburgh in 2000. Therefore, participants did not receive any treatment, including medical, invasive diagnostics, or procedures causing psychological or social discomfort.

### 2.4. Statistical Analysis

All statistical analyses were conducted using the statistical package SPSS 27 for Windows. This software was chosen as it is an efficient and flexible statistical package to conduct analysis for our aims. Several studies, both on this topic and on any other, have used it, e.g., [21,22]. Preliminary analyses included descriptive statistics to describe the group of respondents.

For the three-item scale to measure concern for COVID-19 infection, an explorative factorial analysis was performed to examine its dimensionality, as it was an ad hoc scale for this study.

Subsequently, internal reliability (as Cronbach’s alpha) was calculated for the five validated scales of the SAPH@W instruments and for the ad hoc scale of concern for COVID-19 infection in the workplace, and sum scores by factor were calculated for each scale. This method of factor scores construction was chosen as it preserves the metric of the original scale [23,24], which is more suitable for the subsequent *t*-test analysis.

Preliminary descriptive statistical analyses were conducted. Then, in order to identify the presence of any changes in the feeling of safety in the workplace during the COVID-19 period and in the personal contribution to workplace safety among preschool teachers over two time point, we used a paired samples *t*-test. Specifically, we tested the change of the averages between T0 and T1 with respect to: (1) concern for COVID-19 infection; (2) organization communication; (3) organizational decision-making; (4) situational awareness of contagion risks; (5) fatigue management; (6) personal contribution to workplace safety in relation to COVID-19. Effect size was estimated as Cohen’s d, which is widely used to calculate the effect size for the paired sample *t*-test (e.g., [25]).

Moreover, the relations between the five subscales of the SAPH@W and the scale of concern for COVID-19 infection were analyzed to explore relationships between concern for COVID-19 infection and the perception of safety in the working context.

## 3. Results

### 3.1. Explorative Factorial Analysis for the Scale of Concern for COVID-19 Infection

A maximum likelihood factor analysis was conducted on the ad hoc scale, similar to other studies’ procedure, e.g., [26]. The Kaiser-Meyer-Olkin measure verified the sampling adequacy for the analyses, KMO = 0.59 (“mediocre” according to Hutcheson and Sofroniou [27]). The Kaiser-Meyer-Olkin (KMO) test is widely used as a measure of how suited data is for factor analysis, e.g., [28]. The matrix of correlations showed significant correlations (Table 1) and one extracted factor. The explained variance is 72.72%.

### 3.2. Score at Baseline (T0)

For the first four scales, the 108 respondents indicated an average score of 7.1 (SD 1.81) on a scale ranging from 1 to 10.

For the five dimensions of SAPH@W, the score for fatigue management was around 10 on scale ranging from 5 to 20 (given by the sum of the scores of response scale for the items that compose the scale); the score of decision making, situational awareness of contagion risks and personal contribution to workplace safety in relation to COVID-19 were around 13 on scale ranging 5 to 20. Finally, the score of organizational communication was 15 on scale ranging 5 to 20 (Table 2).

Participants, grouped by their socio-demographic characteristics, did not show deep differences with respect to these dimensions. Specifically, the *t-*test did not show statistically significant differences between teachers who live with husbands and teachers who are single or separated (organization communication: *t* = −1.41; *p* = 0.16; organizational decision-making: *t* = −0.30; *p* = 0.77; situational awareness of contagion risk: *t* = 0.45; *p* = 0.65; fatigue management: *t* = −0.35; *p* = 0.73; personal contribution to workplace safety in relation to COVID-19: *t* = −0.67; *p* = 0.50). Similarly, the *t*-test did not show statistically significant differences between teachers who have children and teachers who have not, regarding SAPH dimensions (organization communication: *t* = 0.58; *p* = 0.56; organizational decision-making: *t* = 1.09; *p* = 0.28; situational awareness of contagion risk: *t* = 0.64; *p* = 0.53; fatigue management: *t* = 1.34; *p* = 0.18; personal contribution to workplace safety in relation to COVID-19: *t* = 0.09; *p* = 0.93). However, teachers who did not have children were more worried both about the differences about the possibility of being infected in the workplace items and the likelihood of being infected with COVID-19 in workplace (“How worried are you about the possibility of being infected in the workplace?”: *t* = −2.31; *p* = 0.02; “How likely are you to be infected with COVID-19 in workplace?”: *t* = −1.99; *p* = 0.04).

### 3.3. Correlations

Through bivariate correlations, we verified the relationship between the concern for COVID-19 infection and the organizational provisions put in place to protect the safety of personnel (Table 3). Examples of papers that used this kind of analysis are [29,30].

Data showed that at T0, the organization’s application of adequate safety measures is related to lower subjective perception of being infected at work: organizational communication, organizational decision-making, situational awareness of contagion risks and fatigue management are inversely correlated with the subjective perception of a contagion risk.

### 3.4. Scores at T0 and at T1: Change over Time

Considering the evolution over time, data showed that the concern for risk of contagion of COVID-19 infection decreased significantly over time. Regarding the SAPH@W dimensions, the only one that showed significant improvement was the personal contribution to workplace safety in relation to COVID-19; the other dimensions showed stability over time (Table 4). Cohen’s *d* showed a medium effect size for the concern for risk of contagion, and small for personal contribution to workplace safety, as an effect size around 0.5 can be considered “medium” and around 0.2 “small”, while around 8.8 is “large” [31,32].

## 4. Discussions

The aim of the study was to describe the workplace concerns and safety perceived by educators, in two successive moments of scholastic year, during the second phase of the pandemic COVID-19, as educators’ perception of safety at work can sustain their wellbeing and therefore allow them to provide better educational service to children.

At T0, the data described subjective perceptions of risk and safety among preschool teachers and perceptions of organizational and personal contributions to workplace safety related to COVID-19 infection. Data were collected during the third wave of the pandemic and before the completion of vaccination of staff of schools of all grades. Regarding the organizational dimensions of workplace safety, the highest scores were found in the dimensions of organizational communication, organizational decision making, situational awareness related to infection risks and personal contribution to workplace safety related to COVID-19. Fatigue management was the most impaired aspect in terms of organizational management of contagion risk. In other words, educators were quite satisfied with the efficiency and effectiveness of organizational communication about COVID-19, the speed, foresight and diligence of the organizational decision-making process related to COVID-19 and situational awareness related to contagion risks and workplace trends. The pandemic situation and the definition of safety protocols to be considered in the school context was a much-discussed topic in the months leading up to the reopening of schools [33], both at the central level (Ministries) and at the local level (within individual schools). It is likely that the issue of workplace safety has never been more important than during this period of the pandemic, and organizations have done their utmost to protect employees as much as possible. This could explain the quite satisfactory results achieved in some dimensions of the SAPH@W. An interesting aspect is the rather high value for the perception of one’s personal contribution to safety at work, indicating a high level of commitment to work. However, there was some dissatisfaction with the institution’s fatigue management. The work of educators and preschool teachers changed significantly during the pandemic: the safety protocol included the use of the mask, physical distance, and reorganization of work [33], which resulted in workers experiencing new and intense work fatigue, both mental and physical, similar to other caregiving occupations considered essential during the COVID-19 pandemic [34,35]. The data indicate that the institution was struggling to comprehend and manage this new mental and physical fatigue. Despite some positive aspects in managing workplace safety, the data at T0 showed high levels of anxiety about COVID-19 infections. The high scores related to anxiety about possible COVID-19 infection were consistent with some previous studies that showed how teachers experienced negative emotional reactions to the COVID-19 pandemic [36,37,38] and highlighted the need for support and intervention [39]. Teachers, particularly preschool teachers, were considered an essential workforce, and they were objectively at higher risk of contagion than other workers who could take advantage of smart working.

At T0, it was also observed how the level of concern for COVID-19 infection was inversely correlated with workplace safety perception, particularly with situational awareness and fatigue management. Working in an organization that was aware of the situation and able to manage fatigue associated with the pandemic was associated with lower levels of fear of infection in the workplace. At T1, our study highlighted how concern about COVID-19 infection decreased over time, while the perception of one’s own contribution to safety increased. The study by Schneider et al. [40] also revealed a change over time in the perception of the risk of infection, which was higher at the beginning of the year and lower in the early summer. One possible explanation could be the seasonality of infections, which tend to decrease at the beginning of summer, leading to a decrease in perceived risk. However, increasing the perception of their own contribution to safety and prevention of COVID-19 led us to believe that educators perceived themselves to be somewhat better able to deal with the COVID-19 situation over time. With this improvement, the perception of the risk of contagion decreased, while dimensions related to organizational safety remained stable. In other words, the pandemic situation between January and May was not objectively very different. In fact, Piedmont was still in the yellow zone with a RT index close to 1, so the risk of infection was still quite high. Nevertheless, teachers were less concerned about COVID-19 contagion in May 2021 than in January 2021. As mentioned in the introduction, preschool teachers had to learn new ways of working (e.g., the “bubble system”) and interacting with children (e.g., using masks). In this regard, this large Municipality was particularly creative, introducing important initiatives such as the “open gardens of public kindergartens” (http://www.comune.torino.it/ucstampa/2020/article_381.shtml (accessed on 1 July 2023)) that allowed children and preschool teachers to meet safely, even during the most difficult weeks of the pandemic. All of these changes were probably a great effort for preschool teachers, but also a great learning process.

## 5. Limitations and Conclusions

This study has some limitations. The sample was small and limited to a specific community. Therefore, our results are not fully generalizable to other similar Italian educational contexts. Moreover, the correlations at T0, which represent a cross-section, do not allow for the testing of cause-effect relationships. Despite these limitations, the present study may be valuable because it focuses on workers’ perceptions of contagion risk in the specific context of educational services, which researchers in Italy largely neglected, even though they were essential. Fear of infection from summer 2021 was likely lower in Italy than in most workers, thanks to the possibility of vaccination against COVID-19 and the presence of vaccinated users in many services. However, users under six years of age in educational services cannot yet be vaccinated or use other protective devices: for workers with this type of user, fear of infection at work could therefore remain high over time. Their perception of safety could thus be lowered, negatively affecting their wellbeing at work and the quality of education provided, if this was not adequately offset by organizational support and interventions. Although it is preliminary, this study may provide some guidance to the management of educational institutions on how to improve the sustainability of educators’ work during high-stress conditions or emergencies as the period of the pandemic was and, at the same time, quality and safety of educational service for children.

Once schools were open again, although teachers observed strict prevention rules (masks, social distancing, etc.), the virus contagion risk was very high. Today, the situation is different. On 5 May 2023, the World Health Organization [41] declared the end of the pandemic. However, this study remains valuable with respect to interesting considering several aspects. Firstly, the pandemic situation has left some “good practices” (in terms of prevention strategies including practicing hand hygiene and social distancing, as well as improving ventilation (according to CDC: www.cdc.gov/coronavirus/2019-ncov/prevent-getting-sick/prevention.html (accessed on 1 July 2023)), in educational contexts that are still being adopted today and that allow educators to contain the biological risk, not only connected to COVID-19, but also to all the other viral forms. Indeed, more than in the years prior to COVID-19, numerous viral forms have spread massively in preschools in recent months. Among these good practices can be cited hand hygiene, smaller work groups, limited access to school and greater attention to symptoms and outdoor play, all less (or not at all) used before the COVID-19 emergency. Secondly, scientists and epidemiologists have highlighted that future pandemics are probably inevitable [18,19]; so, it is very important to have tools and practices that enable successful responses to situations similar to COVID-19. Finally, developing and knowing how to put these actions into practice promotes the perception of possessing NTS [15,17], which has been seen to contribute to workers’ overall perception of safety, and therefore to the perception of offering a service that is attentive to the safety and well-being of users.

## Figures and Tables

**Table 1 children-10-01222-t001:** Matrix correlation between the 3 items of concern about COVID-19 infection.

	1	2	3
1. How worried are you about the possibility of being infected in the workplace?	1	0.496 **	0.857 **
2. How likely are you to be infected with COVID-19 in workplace?		1	0.669 **
3. How worried are you about the possibility of being infected in everyday life?)			1

** correlation is significant at the 0.01 level.

**Table 2 children-10-01222-t002:** Mean and standard deviation with SAPH@W dimension’s score at T0.

Dimensions	N *	M	SD
Organizational communication	106	15.52	3.307
Organizational decision-making	106	13.29	3.950
Situational awareness of contagion risks	107	12.88	4.011
Fatigue management	106	10.31	3.806
Personal contribution to workplace safety in relation to COVID-19	105	13.97	2.751

* N differs between dimensions because cases that left at least one missing answer were excluded.

**Table 3 children-10-01222-t003:** Correlation between the concern for the risk of COVID-19 contagion and the dimensions of the SAPH@W at T0.

	1	2	3	4	5	6
1. Concern for COVID-19 infection	1	−0.071	−0.171	−0.192 *	−0.228 *	0.045
2. Organizational communication		1	0.779 **	0.767 **	0.715 **	0.670 **
3. Organizational decision-making			1	0.854 **	0.724 **	0.617 **
4. Situational awareness of contagion risks				1	0.743 **	0.608 **
5. Fatigue management					1	0.538 **
6. Personal contribution to workplace safety in relation to COVID-19						1

* Correlation is significant at the 0.05 level. ** Correlation is significant at the 0.01 level.

**Table 4 children-10-01222-t004:** Means and standard deviations at T0 and T1 and *t*-test for each dimension investigated.

Item	T	N *	M	SD	t	*p*	*d*
Concern for COVID-19 infection	T0	106	7.10	1.806	5.53	0.000	0.537
	T1	106	6.11	2.053			
Organizational communication	T0	101	15.41	3.332	1.12	NS	-
	T1	101	15.07	3.525			
Organizational decision-making	T0	101	13.23	4.020	0.34	NS	-
	T1	101	13.27	3.728			
Situational awareness of contagion risks	T0	101	12.67	3.993	0.92	NS	-
	T1	101	12.73	3.800			
Fatigue management	T0	101	10.21	3.856	−0.79	NS	-
	T1	101	10.40	3.327			
Personal contribution to workplace safety in relation to COVID-19	T0	101	13.84	2.701	−2.18	0.031	0.228
	T1	101	14.52	2423			

* N differs between dimensions because cases that left at least one missing answer were excluded.

## Data Availability

All data are available by the corresponding author upon reasonable request.

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
