# Peer review of "The Perception of Workplace Safety and of Risk of Contagion among Preschool Teachers during the COVID-19 Pandemic"

_children, 2023, doi:10.3390/children10071222_

Round 1
Reviewer 1 Report
Dear authors,
this is an interesting article that brings new knowledge about ECEC to the field. I have a few minor comments to your text:
Please explain kindergarten and preschool - as these concepts carry different meaning in different countries.
Lines 211 ff Please clarify how the 1-5 scale was transformed to a 5-20 scale.
Lines 256 ff Does “school” here relate to kindergarten and preschools or to schools in general, or both?
Line 323 “Data back to the final phase of the pandemic.” Please clarify what you mean by this sentence.
Additionally, be consistent in your spelling of well-being (lines 107 and 317) compared with wellbeing (lines 58, 92 and 242)
Good luck.
Author Response
Thank you for the suggestion.
The precise answers to each observation are attached.

Reviewer 2 Report
This paper reports on Northern Italian early childhood educators concerns regarding COVID-19 in both January 2021 and May 2021 and compares the responses of both time periods. Its strengths are the thoroughness of the analysis and the generally clear and logical writing style. The weaknesses are that, although the authors point out the importance to these educators of not infecting their families, the authors did not compare the results among those teachers who live with husbands, those who have children, and those who live with an older adult. As well, there are a number of claims made in the Introduction that are without references.
The authors have used the APA style of citation and referencing. As noted in the Instructions for Authors, MDPI journals have their own style of referencing. Please redo all the citations and references to correspond to the style used by Children. Also, with respect to the Instructions for Authors, all sections should be numbered—none of the authors’ sections are numbered and this must be corrected.
Line by line suggested edits.
2 It is preferable to have the title of the work a statement and not include a question. Delete “Offering a safe childhood educational service?”
4 Change “during COVID-19” to “during the COVID-19”.
5 Change “The period COVID-19 had a major impact also on” to “The COVID-19 pandemic had a major impact on”.
8 Change “returning to work in presence” to “returning to in-person work“.
19 Keywords should follow the order in which they appear in the Abstract. Change “preschool context; workplace safety; children safety; COVID-19 pandemic” to “COVID-19; early childhood education; workplace safety; preschool teachers; children; fear of infection”.
23 The authors have mentioned “Several studies”, yet only one study is cited. Please cite more than one study if they authors want to say there are
39 Change “vaccination COVID-19” to “vaccination for COVID-19”.
41 Change “this potential risk of infection changed” to “did this potential risk of infection change”.
51 “Dai et al., 2019” is not part of the reference list. As well, it is likely not possible that a paper published in 2019 would be a study of healthcare workers fears during the pandemic as the pandemic only began in late November 2019. Please check your reference.
58 Again, “Dai et al., 2019” is not in the reference list.
59 Change “to provide” to “in providing”.
61-83 There are a number of claims made in both of these paragraphs, yet neither paragraph contains citations to references supporting these claims. References must be provided for each sentence that makes a claim.
112 In the Instructions for Authors available on the Children website, authors are asked to, “Finally, briefly mention the main aim of the work and highlight the main conclusions.” Although the authors have presented the main aim of the work, they have not highlighted the main conclusions. They are asked to please do so at the end of the Introduction.
115-132 The authors have not indicated how the participants were selected and how they obtained the email addresses of the potential participants. As well, the authors are asked to indicate if any incentive was provided for participation. Finally, the authors are asked to state how informed consent was obtained.
176 The authors are asked to state how many of the staff responded to only the first questionnaire and to indicate the dropout rate with respect to the second questionnaire as a percentage.
178 Given that an important aspect of this study was the concern that the early childhood educators had regarding infecting their own families, data should be available on the number of preschool teachers who were married, the number who had children and the number who lived with an elderly relative. Was this information collected? If so, it should be reported. If not, it should have been collected and analyzed.
180 Please explain why SPSS 27 was selected for statistical analyses and provide a recent peer reviewed reference to demonstrated it has been used for other similar COVID-19 studies.
182 Please explain why it was important to examine the dimensionality of the concern for COVID-19 infection and provide a peer reviewed reference to show that this has been done in other COVID-19 studies. As well, remove the italics from “ad hoc” as this is the style for MDPI journals.
196-198 Please explain why the effect size was estimated by Cohen’s d and why the relations between the five subscales of the SAPH@W and the scale of concern for COVID-19 infection were analyzed and provided peer reviewed research to demonstrate that these methods have been used in relation to other similar COVID-19 studies.
201 Remove the italics from “ad hoc” as this is the style for MDPI journals.
202 Change “sapling” to “sampling”. Please explain why the Kaiser-Meyer-Olkin measure was used to verify the sampling adequacy for the analyses and find a current reference to support the use of this measure for studies related to COVID-19.
205 The title should indicate that Table 1 relates to change of the averages between T0 and T1.
205-206 Please single space the rows, similar to Table 4.
209 In line 176, the authors stated that there were 108 participants. Here they state there are 107. Please correct whichever of the two is incorrect.
216 The title of Table 2 should specify that it relates to the score at baseline.
216-217 Please explain why the N differed for “Situational awareness of contagion risks” and for “Personal contribution to workplace safety in relation to COVID-19” and why none of the N = 108. As well, please single space the rows, similar to Table 4.
227-228 Please single space the rows, similar to Table 4.
237-238 The authors are asked to explain why the N for “Concern for COVID-19 infection” was 106 and the N for all other items was 101.
301-302 Provide a reference to “open gardens of public kindergartens”.
307 Change “other Italian” to “other similar Italian”.
323 “Data back to the final phase of the pandemic.”—the meaning of this sentence is unclear. Please rephrase.
324 Provide a reference to the WHO declaration and, as this is the first time the WHO is mentioned in the paper, please spell it out in full.
326 Please define what the authors mean by “good practices”.
333 Change “hiving” to “having”.
Suggested changes to the English have been made in the line by line suggested edits above.
Author Response

(The authors gave the same response as above.)

Round 2
Reviewer 2 Report
Thank you to the authors for the detailed cover letter they provided. The responses to each of the points have addressed this reviewer’s concerns.
However, when the reviewer asked for clarification on particular matters and references, these were intended to be added to the text as well, not just part of the cover letter. Furthermore, it appears that the author who composed the cover letter, which is written in very clear English, was not the same person who made the corrections to the text. The English in the text itself has required suggested modifications in order to be correct. As well, new references added by the author who made the corrections to the text were added in the APA style of citation rather than the MDPI style of numbering. These will need to be standardized to the preferred style.
Thank you for adding the numbering to the sections and subsections. Please also include a period after the final digit of the subsections numbering. As well, do not include a space between citations if more than one reference is cited at a time.
65 Please number this citation and put the reference in the list of references.
70 Please number this citation and put the reference in the list of references.
120 Change “Even if the pandemic is now ended having tools and practices that enable to face situations similar to COVID-19, and in general supporting the development of the NTS to enhance self- efficacy perception and wellbeing- can be important, in the case of future, possible new pandemic” to “Although the COVID-19 pandemic is now ended, having tools and practices that enable successful work-related responses as well as supporting the development of NTS to enhance self-efficacy perception and wellbeing can be important in the case of possible new pandemics”.
125-193 Please reorganize these sections so that they follow the timeline of the study. As such, move the “Measures” subsection to be the first, the “Participants” subsection to be second, and the “Data collection” to be third.
145 Delete “some”.
186-187 Change “The participants were intercepted during an on-line training course. Here the subscripts to the study and the email addresses have been collected” to “The participants were identified during an on-line training course when they were asked to participate in the study and their email addresses were collected.” Were there any participants in the on-line training course who refused to be part of the study? If so, this needs to be mentioned.
197 The referenced provided in the authors’ cover letter regarding SPSS 27 are to be cited here, not just in the cover letter.
200 “As this scale was developed ad hoc for this study we performed explorative analysis to consider its unidimensional ad successively we analyzed it internal consistency by Cronbach’s alpha. A similar procedure was followed in the first part of this paper, e.g.:
Zenker, S., Braun, E., & Gyimothy, S. (2021). Too afraid to travel? Development of a pandemic (COVID-19) anxiety travel scale (PATS). Tourism Management, 84, 104286. https://doi.org/10.1016/j.tourman.2021.104286”
Please include this information provided in the authors’ cover letter in the text and cite the reference provided.
216 “Cohen’s d is used widely for calculating the effect size for the paired sample T-test. An example can be found in this work:
Yanto, D. T. P., Kabatiah, M., Zaswita, H., Jalinus, N., & Refdinal, R. (2022). Virtual Laboratory as A New Educational Trend Post Covid-19: An Effectiveness Study. Mimbar Ilmu, 27(3), 501–510. https://doi.org/10.23887/mi.v27i3.53996
The relations between the five subscales of the SAPH@W and the scale of concern for COVID-19 infection were analyzed in order to explore relationships between concern for COVID-19 infection and the perception of safety in the working context, as we added in the manuscript. An example of paper that used this kind of analysis is:
Labrague, L. J., & de Los Santos, J. A. A. (2021). Fear of Covid‐19, psychological distress, work satisfaction and turnover intention among frontline nurses. Journal of nursing management, 29(3), 395-403.”
Please include this information provided in the authors’ cover letter in the text and cite the references provided.
The cover letter also provides this reference that follows. However, it is unclear what claim this reference is intended to support. Please cite this reference where appropriate:
“MacIntyre, P. D., Gregersen, T., & Mercer, S. (2020). Language teachers’ coping strategies during the Covid-19 conversion to online teaching: Correlations with stress, wellbeing and negative emotions. System, 94, 102352.”
219 “The Kaiser-Meyer-Olkin (KMO) Test is widely used as measure of how suited data is for Factor Analysis.
An example of paper that uses it is:
Singh, Khundrakpam Devananda. Coronavirus Anxiety Scale: A Validation Study in an Indian Population. Medical Journal of Dr. D.Y. Patil Vidyapeeth 14(3):p 303-307, May–Jun 2021. | DOI: 10.4103/mjdrdypu.mjdrdypu_504_20”
Please include this information provided in the authors’ cover letter in the text and cite the references provided.
236-237 Change “differences respect” to “differences with respect”.
336 Please cite this reference with the appropriate number and include it in the list of references.
358-361 Change “Data back to the final phase of the pandemic, when schools were open again, teachers had to observe strict prevention rules (masks, social distancing, etc.) and the virus contagion was risk was very high. Today the situation is different and on 5 May 2023 the World Health Organization [33] declared the end of the pandemic” to “Once schools were open again, although teachers observed strict prevention rules (masks, social distancing, etc.), the virus contagion was risk was very high. Today, the situation is different. On 5 May 2023 the World Health Organization [33] declared the end of the pandemic”
361-362 Change “can still be considered interesting considering” to “remains valuable with respect to”.
Please include this information provided in the authors’ cover letter in the text and cite the references provided. However, change to “Good practices in terms of prevention strategies include practicing hand hygiene and social distancing as well as improving ventilation (according to CDC: www.cdc.gov/coronavirus/2019-ncov/prevent-getting-sick/prevention.html).”
369-371 Change “scientist and epidemiologist highlights that future pandemics are probably inevitable [17; 18] so it is very important having tools and practices that enable to face new situations like COVID-19” to “scientists and epidemiologist have highlighted that future pandemics are probably inevitable [17,18]; so it is very important to have tools and practices that enable successful responses to situations similar to COVID-19”.
372-373 Change “the NTS [15, 16], which, it has been seen, contributes” to “NTS [15,16], which have been seen to contribute”.
The new additions that have been made to the revision require changes to the English. Changes have been suggested in the appropriate lines in the Comments and Suggestions for Authors.
Author Response
Thank you, the answers are attacched.
